

# Grazing exclosures solely are not the best methods for sustaining alpine grasslands

Xixi Yao[1], Jianping Wu[1,2], Xuyin Gong[2], Xia Lang[2] and Cailian Wang[2]

[1] College of Animal Science and Technology, Gansu Agricultural University, Lanzhou, People's Republic of China
[2] Gansu Academy of Agricultural Science, Lanzhou, People's Republic of China

## ABSTRACT

**Background**. Grazing is widely regarded as a critical factor affecting the vegetation structure, productivity and nutritional value of natural grasslands. To protect and restore degraded grasslands, non-grazed exclosures are considered as a valuable tool. However, it is not clear whether long term non-grazed exclosures of grazers can improve the condition and nutritional value of vegetation and soil properties.

**Methods**. We have compared the impact of long-term non-grazed and continuous grazed management strategy on vegetation structure, nutritional values and soil properties of alpine meadow of the Qinghai-Tibet Plateau by field investigation (11–13 years) and indoor analysis during 2015–2017.

**Results**. Our results showed that long-term non-grazed exclosures clearly increased the aboveground biomass and coverage of plant functional types. Long-term non-grazed exclosures improved the development of all vegetation types, except NG (GG, grass species type; SG, sedge species type; LG, leguminous species type; FG, forbs species type and NG, noxious species type). Long-term non-grazed exclosures significantly improved all six measured soil properties (TN, total nitrogen; TP, total phosphorus; TK, total potassium; AN, available nitrogen; AP, available phosphorus and AK, available potassium) in 0–10 cm soil layer, considerable effect on the improvement of all measured soil properties, except TK in 10–20 cm soil layer and all measured soil properties, except TN and TK in 20–30 cm soil layer were observed. However, long-term non-grazed exclosures significantly decreased biodiversity indicators i.e., species richness, Shannon diversity index and Evenness index of vegetation. A substantial decrease in the density, biodiversity and nutritional values (CP (crude protein), IVTD (*in vitro* ture digestibility) and NDF (neutral detergent fiber)) of all vegetation types, except NG were recorded. While a downward trend in aboveground biomass and all measured soil properties except TP and TK were observed during 2015–2017 in alpine meadows due to long-term grazed treatment. The density, diversity and nutritional value (CP and IVTD) of long-term non-grazed alpine meadows showed a downward trend over time (2015–2017). By considering the biodiversity conservation and grassland livestock production, long-term non-grazed exclosures are not beneficial for the improvement of density, biodiversity and nutritional values of plant functional types. Thus, our study suggests that rotational non-grazed and grazed treatment would be a good management strategy to restore and improve the biodiversity and nutritional values of plant functional types in natural grassland ecosystems.

Corresponding author
Jianping Wu, 18993075304@163.com

# INTRODUCTION

Grasslands occupy 40% of the earth's land surface and play an important role in ecosystem functions and grassland animal husbandry (*Kemp et al., 2013*; *Jing et al., 2014*; *Wesche et al., 2016*; *Török et al., 2016*; *Török & Dengler, 2018*). China occupies 400 million hectares of grasslands, which account for 42% of the world's land area, out of which 240 million hectares are located in the Qinghai-Tibet Plateau (QTP), supporting 16 million people directly (*Kemp et al., 2013*). About 70% of China's population resides in rural areas, and many of these people rely on grassland animal husbandry (*Yang et al., 2012*). Therefore, QTP is an important source for the survival and development of people in China and also a significant ecological barrier shaping genetic structure (*Kemp et al., 2013*). Alpine meadows cover almost 85% area of the QTP and play an essential role in ecosystem function and grassland animal husbandry in china (*Jiang et al., 2012*; *Wen et al., 2018*).

Alpine meadows have been gradually degrading and desertifying since 1980. Previous studies reported that 90% of alpine meadows in QTP are degraded and 35% of this area is described as "black-soil-type alpine meadow" due to the severity of degradation (*Dong et al., 2007*; *Li et al., 2017*). Grassland degradation may be due to many reasons. However, increasing population pressure, livestock quantity and overgrazing are usually considered as the main reasons for grassland degradation (*Kemp et al., 2013*; *Chen et al., 2014*). Overgrazing may lead to significant changes in plant community composition and structure (*Zhou et al., 2005*). In addition, overgrazing leads to an increase in potential evapotranspiration and local global warming and further accelerates degradation of alpine meadows (*Du et al., 2004*). Degradation of grasslands due to overgrazing will start a vicious circle in which degraded grasslands will be degraded due to invasions of rodents (*Kang et al., 2007*).

In order to relieve the problem of grassland degradation in QTP, the Chinese government has initiated a project at local state and authorities in 2004 named the Returning Grazing Land to Grassland Project (RGLGP). As a management tool of this project, exclosures was extensively used to to protect and restore degraded grassland ecosystems all over the world in recent decades (*Wu et al., 2009*; *Jing et al., 2014*; *Cheng et al., 2016*). This strategy has been in consideration for more than a decade in degraded and overgrazed areas of QTP, revealing a question: is this strategy successful in restoring degraded alpine meadows?

The degradation of grasslands has attracted great attention in recent years and stimulated a large number of studies on the use of exclosures (*Wu et al., 2009*; *Wei et al., 2012*; *Shi et al., 2013*). Studies showed that exclosures increase aboveground biomass, coverage, species diversity and soil nutrient content (*Jiao, Wen & An, 2011*; *Shi et al., 2013*; *Li et al., 2017*; *Wen et al., 2018*) but decrease species density, richness and biodiversity by diminishing the dominant competitor species present during grazing (*Mayer et al., 2009*; *Shi et al., 2013*). Exclosures also increased the nutritional value of forage (*Schönbach et al., 2012*; *Ren et al.,*

*2016*), which in turn affected the livestock production and performance (*Mysterud et al., 2001*; *Mysterud et al., 2011*). However, exclosures can lead to wastage of natural resources in livestock production (*Cuevas & Le Quesne, 2005*). Thus, specific research should be done for the proper management of ecosystems and the achievement of protection objectives. In QTP, much research has been done to explore the effects of exclosures on alpine meadow ecosystem, vegetation structure, vegetation succession and soil characteristics under different degradation gradient, grazing intensities and grazing regime (*Pettit, Froend & Ladd, 1995*; *Gibson et al., 2000*; *Li et al., 2017*; *Wen et al., 2018*). However, less work has been done to study the effects of exclosures on nutritional values of plant functional types. In fact, nutritional values of vegetation are also of great importance for animal production along with ecosystem functions and services (*Wen et al., 2018*; *Shang et al., 2013*).

The livestock industry in QTP accounts for a significant proportion of government income (*Kang et al., 2007*). Livestock production is usually restricted by herbage nutritional yield, which depends on aboveground net primary productivity (ANPP) and herbage nutritional values (*Ren et al., 2016*). Forage with high nutritional values is characterized by high concentration of crude protein (CP), *in vitro* true digestibility (IVTD) and low concentration of neutral detergent fiber (NDF). At present, most areas of the grassland are in some state of degradation and the ANPP of grassland has decreased (*Li et al., 2017*; *Wen et al., 2018*). Meanwhile, the number of livestock in QTP region is still increasing, causing more overgrazing, resulting more grassland degradation and reduced ANPP of grassland (*Yang et al., 2012*). In addition, climatic factors also influence the grassland, such as extremely low temperature diminishes grass growth from October to May and due to low herbage mass animal productivity is severely limited (*Yang et al., 2012*). So, the importance of forage nutritional values in the production of livestock is highly recognized. Little work has been done focusing the effects of grassland degradation on vegetation characteristics, vegetation nutritional values and soil properties; in particular, the characteristics of plant functional types and the nutritional values of plant functional types as a whole have not yet been reported. The effects of long-term non-grazed and grazed management strategy on vegetation characteristics, vegetation nutritional values and soil properties in grassland are unclear.

In order to better understand the restoration and management of degraded grassland in QTP, it is necessary to study the vegetation characteristics, nutritional values and soil properties of alpine meadows as a whole. We hypothesized that aboveground biomass, coverage, density, biodiversity, nutritional values of vegetation and soil properties will be improved in the absence of grazed due to the absence of disturbance from herbivorous livestock. Thus, in this study, we compared characteristics of plant functional types, functional types nutritional values and soil properties during long-term non-grazed and grazed alpine meadows of QTP in order to evaluate whether long-term non-grazed exclosures can improve the condition and nutritional values of plant functional types and soil properties. The assessment of long-term non-grazed exclosures as management strategy will assist in preventing negative impacts on ecosystem and full utilization of grasslands resources.
## MATERIAL AND METHODS

### Study site

The study area was located on the Sanding Village in the northeastern edge of the Qinghai-Tibet Plateau, Kangle Town, Sunan County, Zhangye City of Gansu Province, in China (99°48′E, 38°45′N, and 3,200 m above sea level). The annual average precipitation was 255 mm (1985–2017), with ∼85% occurring during the growing season (May–September) (Fig. 1B). The average annual temperature was approximately 3.8 °C (1985–2017) (Fig. 1A). The annual cumulative temperature ($\geq$0 °C) was approximately 2323.9 °C (1985–2017) (Fig. 1C). The vegetation growth period was from June to September i.e., approximately 4 months. The type of grassland belonged to alpine meadow. The species richness was high, with 12–24 species per $m^2$ in this vegetation meadow. The vegetation was divided into five classes that included (gramineous grasses (GG); sedge grasses (SG); leguminous grasses (LG); forbs grasses (FG) and noxious species (NG)). The dominant species included: *Kobresia humilis*, *Polygonum viviparum*, *Potentilla fruticosa* and *Caragana sinica* (*Yang et al., 2012*) (Species are shown in Table 1). The soil of the alpine meadow belonged to meadow soil with high calcium content (*Kemp & Michalk, 2011*).

### Experimental design

The study area, including all research sites, was slightly degraded because it was used as summer pasture and mainly grazed from June to September by Gansu Alpine Fine-wool sheep before 2005. Two treatments, i.e., non-grazed and grazed treatments, were started in 20th June, 2005 under the same conditions, while sample collection was started from 2015. In each treatment three sites, each site occupied 4 ha area (approximately 100 m away from each other, to insure an unification of experiment conditions, all sites had similar slope gradient, aspect, elevation and soil type (*Cheng et al., 2016*; *Li et al., 2017*; *Wen et al., 2018*)), were randomly selected over a homogeneous area (total 24 ha area). At each site, using the line transects method, three 100 m ×100 m monitoring blocks (almost at a distance of 50 m) with the same conditions were selected, altogether eighteen sample blocks for two treatments were chosen. The non-grazed sites had been excluded from livestock grazing for 10 years. The grazed sites were fenced for the whole year and were freely grazed from 20th June to 20th August with a 4.5 livestock units/ha (Gansu Alpine Fine-wool sheep, 18-month-old female sheep, average 35 kg live weight) during the grazing period. The grazing experiment started on 20th June, 2005. Animal welfare and experimental procedures were carried out in accordance with the Guide for the Care and Use of Laboratory Animals, Ministry of Science and Technology of China and were approved by College of Animal Science and Technology of Gansu Agricultural University. Every effort was made to minimize animal pain, suffering and distress and to reduce the number of animals used. Gansu Alpine Fine-wool sheep management followed traditional practice, in which grazing sheep were kept in the grazing sites day and night with freely available drinking water. The paddock was owned by a local farmer, who agreed to its use for this experimental study. There were no endangered or protected species within the paddock.

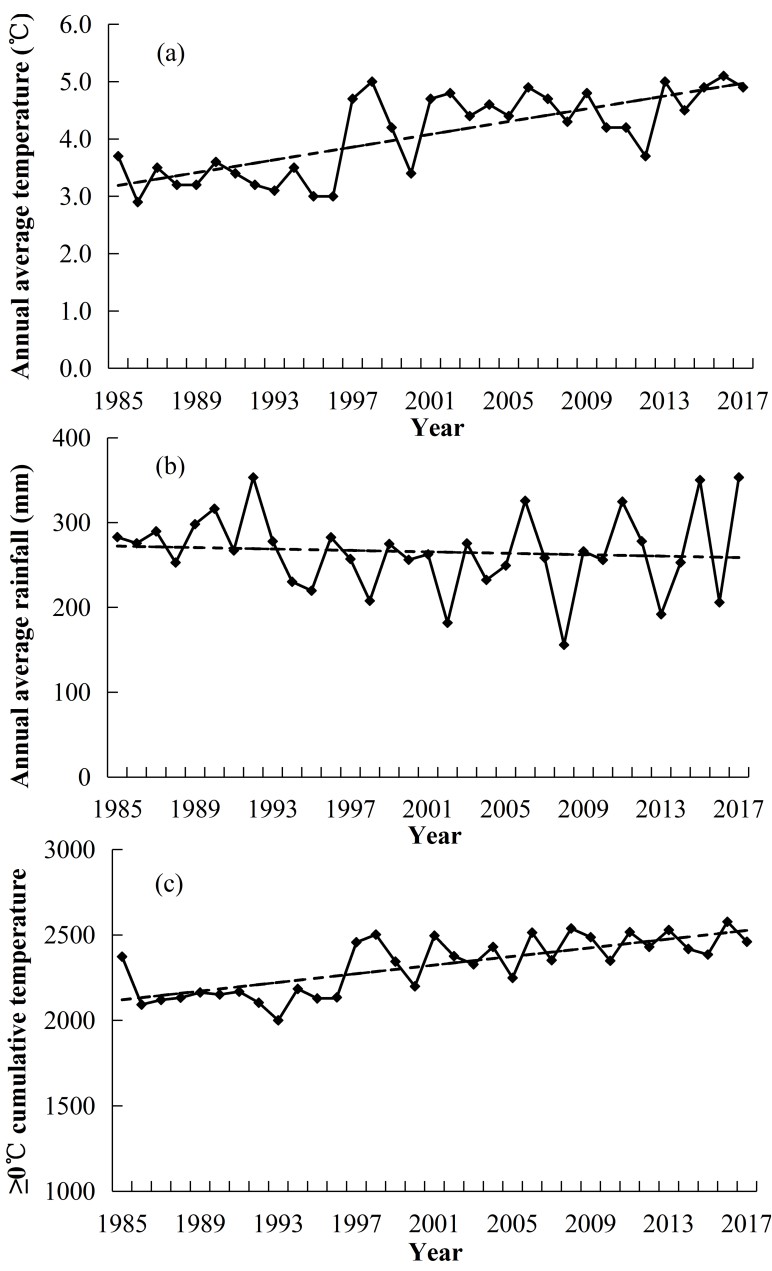

**Figure 1  Changes in annual average temperature (°C) (A), annual average precipitation (mm) (B) and cumulative temperature (≥0 °C) (C) in the region of this study.**

## Aboveground vegetation community survey, plant and soil sampling

Successive samples were collected on mid-August of 2015, 2016 and 2017 when aboveground biomass at the peak of biomass production. In every experimental block, five random sampling of quadrats (1 m × 1 m) were done, having distance between each quadrat over 1 m from edge to edge to eliminate the effect of margin. Species composition, life type, edibility, coverage, aboveground biomass and density were recorded in each quadrat and
**Table 1** The mean aboveground biomass (g/m$^2$) for species present at surveyed quadrats of long-term non-grazed and grazed alpine meadows in 2015, 2016 and 2017.

| Species | Life type | Edibility | Functional types | Aboveground vegetation productivity (g/m$^2$) | | | | | |
|---|---|---|---|---|---|---|---|---|---|
| | | | | 2015 | | 2016 | | 2017 | |
| | | | | Non-grazed | Grazed | Non-grazed | Grazed | Non-grazed | Grazed |
| *Elymus nutans* Griseb. | P | E | GG | 38.72 | 12.13 | 28.21 | 8.23 | 35.11 | 3.15 |
| *B. Sylvaticum* (Huds) Beauv | P | E | GG | 17.22 | 7.86 | 6.15 | 2.18 | – | 2.44 |
| *Roegneria kamoji* Ohwi | P | E | GG | – | 1.22 | – | 0.52 | – | – |
| *Stipa aliena* Keng | P | E | GG | 20.15 | 5.83 | 17.86 | 7.89 | 26.25 | 9.51 |
| *Poa crymophila* Keng | P | E | GG | 54.90 | 20.05 | 45.99 | 13.51 | 49.86 | 17.86 |
| *Kobresia humilis* (C. A. Mey. ex Trautv.) Sergiev | P | E | SG | 85.32 | 27.15 | 58.16 | 27.21 | 68.55 | 23.81 |
| *Scirpus triqueter* L. | P | E | SG | – | 8.06 | – | 6.35 | – | – |
| *Carex capillifolia* (Decne.) Zhang (2015) | P | E | SG | 65.72 | 17.05 | 67.30 | 13.50 | 59.38 | 14.58 |
| *Carex melanocephala* Turcz. ex Bess. | P | E | SG | 20.01 | 6.15 | 11.86 | 2.67 | 24.12 | 2.50 |
| *Caragana sinica* (Buc'hoz) Rehder | P | E | LG | 42.35 | 38.11 | 30.50 | 28.85 | 43.50 | 20.36 |
| *Medicago ruthenica* var. inschanica | P | E | LG | 26.50 | 24.23 | 6.98 | 6.86 | 10.42 | 3.05 |
| *Gueldenstaedtia verna* (Georgi) Boriss | P | E | LG | – | 2.48 | – | 1.21 | – | – |
| *Medicago ruthenica* (L.) Trautv. | P | E | LG | 2.45 | 1.24 | 4.20 | 2.21 | – | 0.45 |
| *Polygonum viviparum* L. | P | E | FG | 12.30 | 9.51 | 21.22 | 18.77 | 20.56 | 20.30 |
| *Polygonum macrophyllum* D. Don | P | E | FG | 7.45 | 6.80 | 9.70 | 5.06 | 15.83 | 13.05 |
| *Saussurea japonica* (Thunb.) DC. | P | E | FG | 4.50 | 2.55 | – | 3.45 | – | – |
| *Potentilla fruticosa* L. | P | E | FG | 17.02 | 12.90 | 24.11 | 15.01 | 34.50 | 7.01 |
| *Salix oritrepha* Schneid. | P | E | FG | 5.51 | 4.15 | 7.51 | 4.92 | – | 0.21 |
| *Heteropappus altaicus* (Willd) Novopokr | A | E | FG | – | 8.05 | – | – | – | – |
| *Thalictrum aquilegifolium* | P | I | NG | 3.72 | 3.26 | 2.89 | 2.45 | – | 3.15 |
| *Taraxacum mongolicum* Hand.-Mazz. | P | I | NG | – | 1.25 | – | 1.88 | – | 2.21 |
| *Leontopodium alpinum* | P | I | NG | 2.58 | 1.98 | – | 1.56 | 2.65 | – |
| *Potentilla bifurca* Linn. | P | I | NG | 3.10 | 2.77 | 4.58 | 3.51 | 3.82 | 4.68 |
| *Stellera chamaejasme* L. | P | I | NG | 1.32 | 2.12 | 3.66 | 2.66 | 2.52 | 6.85 |
| *Oxytropis ochrocephala* | P | I | NG | 4.58 | 3.54 | – | – | 1.52 | – |
| *Geranium wilfordii* Maxim. | P | I | NG | – | – | – | 0.85 | – | – |
| *Oxytropis kansuensis* | P | I | NG | 3.85 | 1.99 | 7.89 | 6.08 | 5.77 | 5.27 |
| *Gentiana scabra* Bunge | A | I | NG | 0.65 | 0.81 | 4.56 | 6.50 | 6.25 | 2.80 |

**Table 1** (*continued*)

| Species | Life type | Edibility | Functional types | Aboveground vegetation productivity (g/m²) | | | | | |
|---|---|---|---|---|---|---|---|---|---|
| | | | | **2015** | | **2016** | | **2017** | |
| | | | | Non-grazed | Grazed | Non-grazed | Grazed | Non-grazed | Grazed |
| *Pedicularis ikomai* Sasaki | P | I | NG | 1.21 | 1.13 | – | – | – | 2.11 |
| *Gentianopsis paludosa* | A | I | NG | – | 0.85 | – | 1.51 | 1.86 | 2.98 |
| *Polygonum sibiricum* Laxm. | P | I | NG | – | 1.67 | – | 2.21 | – | – |
| *Plantago asiatica* L. | P | I | NG | 0.41 | 2.21 | – | – | – | 2.25 |
| *Artemisia hedinii* Ostenf. et Pauls. | A | I | NG | – | 1.15 | 3.64 | – | – | 3.66 |
| *Gentiana macrophylla* | P | I | NG | 0.35 | 0.68 | 1.68 | 1.61 | 1.89 | 1.64 |
| *Rheum pumilum* Maxim. | P | I | NG | – | | 0.84 | 0.26 | – | 0.21 |

**Notes.**
For life types of species, P and A represents perennials and annuals respectly. For edibility, E and I represents edible and inedible respectively. For five functional types, GG, SG, LG, FG and NG represents grass species type, sedge species type, leguminous species type, forbs species type and noxious species type respectly. "–", not present.

each plant species was clipped to 1-cm stubble height. Additionally, the plant functional types were divided into five classes that included GG (grass species type), SG (sedge species type), LG (leguminous species type), FG (forbs species type) and NG (noxious species type) (*Wu et al., 2009*; *Zhang et al., 2018*). Edible plants species mean those plants that can be eaten by animals while noxious species mean noxious plants. Noxious species means plant species that are classified as undesirable, noxious, exotic, injurious, or poisonous, pursuant to local government law, which are of foreign origin, and can directly or indirectly injure crops, other useful plants, livestock, or poultry (*Parker, 1949*). Dry matter in each functional group under the constant weight of every quadrat drying at 70 °C for 48 h was determined. A total of 270 quadrants were recorded during 3 years of experiment for long-term non-grazed and grazed treatments. Total coverage, aboveground biomass (dry matter) and species density of alpine meadow plant functional types were recorded. The information of all species is shown in Table 1. Species abundance was calculated by using the number of species in each square. Species richness ($S$), Shannon diversity index ($H$) and Evenness index ($E$) of the vegetation was calculated using the formula:

$$\text{Species richness } (R): R = S$$

$$\text{Shannon diversity index } (H): H = -\sum_{i=1}^{s}(Pi \ln Pi)$$

$$\text{Evenness index } (E): E = \frac{H}{\ln S}.$$

Where $S$, $H$, $P_i$ represents total species of alpine meadow vegetation community, Shannon diversity index and density proportion of i species, respectively (*Shannon, 1948*).

Along with this, surface soil was also removed from five spots of each quadrat. The center and four diagonal corners of each sampling quadrat were selected using the earth boring auger in three soil layers: 0–10, 10–20 and 20–30 cm. Soil samples from five spots of each quadrat were mixed together to use as one soil sample. There were 270 soil samples in

total from same soil layer for long-term non-grazed and grazed treatments during 3 years of experiments.

## Nutrients chemical analysis

Nutritional values of four edible functional types were evaluated. Plant samples were dried, crushed and passed through 1 mm mesh screening by using Foss Tecator Cyclotec 1,093 sample mill. The plant nutrition parameters included crude protein (CP), neutral detergent fiber (NDF) and *in vitro* ture digestibility (IVTD). Plant nitrogen (N) concentration was measured using Foss fully automated Kheltec 8400 (*Feldsine, Abeyta & Andrews, 2002*) and utilized to calculated crude protein (CP) by using formula: % CP = % N ×6.25. NDF was analyzed using an ANKOM 2000 Fiber Analyzer on the basis of the two-stage method (*Tilley & Terry, 1963*) while *in vitro* true digestibility (IVTD) was determined using an Ankom F57 filter bag (*Goering & Van Soest, 1970*; *Van Soest, Robertson & Lewis, 1991*).

Soil samples were passed through 0.14 mm mesh screening after air-drying. The measurement of soil samples according to methods (*Miller & Keeney, 1982*). Total nitrogen (TN) was obtained by the semi-micro Kjeldahl method. Total phosphorus (TP) and total potassium (TK) were determined with an inductive coupled plasma (ICP) emission spectrometer after digestion of the samples in concentrated $HNO_3$. Available nitrogen (AN) was determined using the continuous alkali-hydrolyzed reduction diffusion method. Available phosphorus (AP) was extracted with sodium bicarbonate and determined by Olsen method. Available potassium (AK) was determined by $H_2SO_4$-$HCLO_4$ digestion and the molybdenum antimony-ascorbic acid colorimetric method.

## Data analysis

Mixed Model option in SPSS version 19.0 (IBM Corp., Armonk, New York, USA) based on an autoregressive covariance structure through ANOVA analyzing data was used. There were 270 observations [2 treatments ×9 blocks ×5 quadrants ×3 years] for each functional type variables (Coverage, Plant density, Aboveground biomass, CP, IVTD and NDF) and soil property variables in each soil depth (Total N, Total P, Total K, Available N, Available P and Available K). Repeated measurement analysis for each functional type variables and soil property variable in every soil depth were performed using a mixed-effects model, including grazing treatment (non-grazed, grazed), and blocks were selected to study the fixed effects with year (2015, 2016, 2017) as a repeated effect and their interactions. The ANOVA analysis was followed by least significant difference (LSD) tests ($P < 0.05$).

## RESULTS

### Vegetation characteristics response to non-grazed treatment and year variations

The vegetation coverage and aboveground biomass of non-grazed treatment were significantly higher than grazed treatment (Fig. 2; Table 2). The vegetation density, Species richness (*R*), Shannon diversity index (*S*) and evenness index (*E*) of non-grazed treatment were considerably lower than grazed treatment (Fig. 2; Table 2). The sampling year notably influenced the vegetation coverage and density (Fig. 2; Table 2). The effect
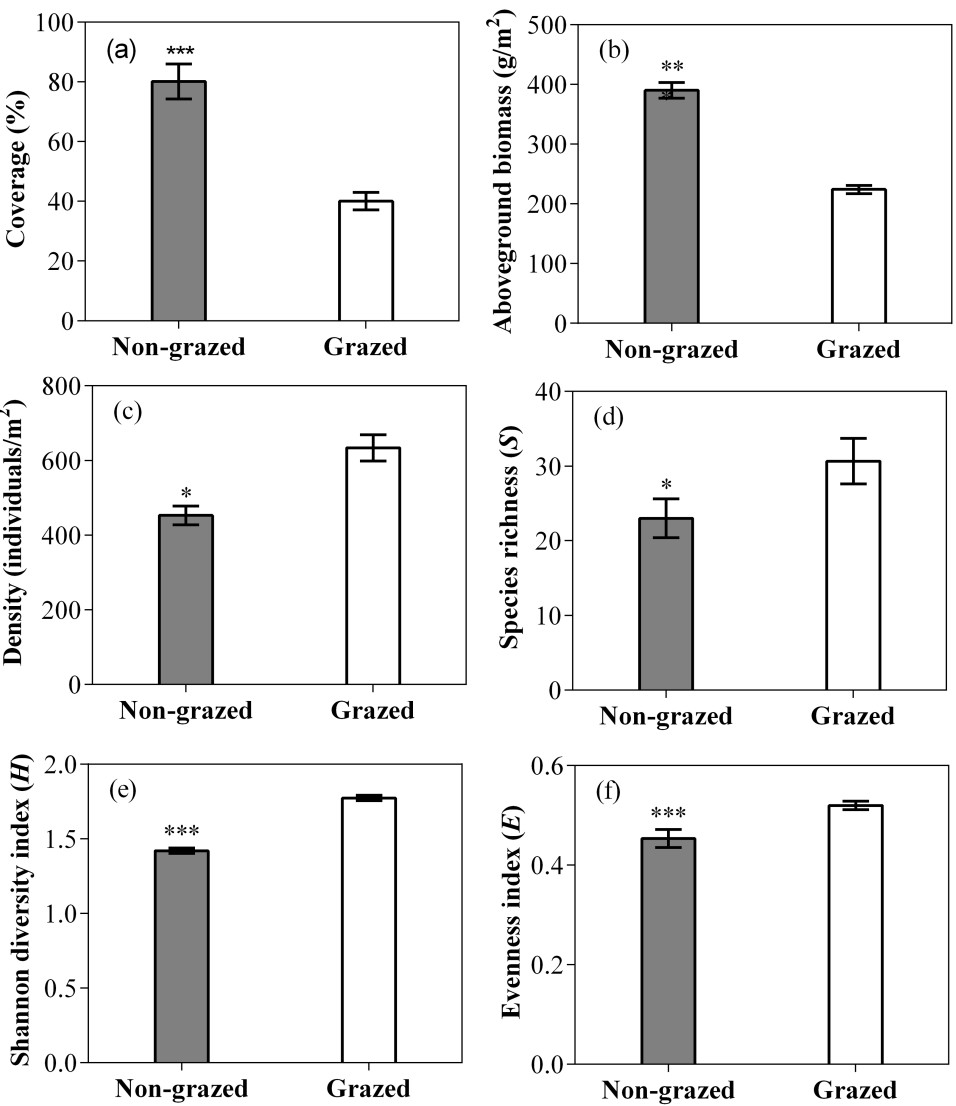

**Figure 2** **Effect of long-term non-grazed and grazed on vegetation coverage (A), aboveground biomass (B), density (C), richness index (D), Shannon diversity index (E) and Evenness index (F) of alpine meadows.** The values (Mean ± SE) are means of 3 years (2015, 2016 and 2017); The symbols represent signigicant difference between non-grazed and grazed alpine meadow treatments, \*\*\*$P < 0.001$, \*\*$P < 0.01$, \*$P < 0.05$; ns, no significant difference.

of non-grazed treatment on the coverage and density were highly influenced by sampling year (Fig. 3; Table 2).

In comparison to grazed treatment, non-grazed treatment considerably increased the coverage of all five vegetation types (Fig. 3; Table 2). Significant increase in the aboveground biomass of GG, SG was observed during non-grazed alpine meadows. Non-grazed treatment also increased the density of all five vegetation types (Fig. 3; Table 2). The sampling year only increased the coverage of LG and NG. The sampling year also increased the density of all vegetation types, except FG. However, significant effects of interaction between

**Table 2** The effects of years (2015, 2016 and 2017), non-grazed (compared with grazed) and interaction between non-grazed and year coverage (a, %), aboveground biomass (b), density (c) for total vegetation and five functional types, GG, SG, LG, FG and NG represents grass species type, sedge species type, leguminous species type, forbs species type and noxious species type respectly.

| Functional types | Items | P-values of variables | | | | |
|---|---|---|---|---|---|---|
| | | Non -grazed | Plots | Year | Non-grazed × Year | Year × Plots |
| Total | Coverage | 0.001 | 0.725 | 0.014 | 0.925 | 0.915 |
| | Plant density | 0.006 | 0.684 | 0.027 | 0.849 | 0.955 |
| | Aboveground biomass | 0.012 | 0.551 | 0.326 | 0.001 | 0.924 |
| GG | Coverage | 0.001 | 0.594 | 0.106 | 0.582 | 0.842 |
| | Plant density | 0.003 | 0.421 | 0.013 | 0.924 | 0.882 |
| | Aboveground biomass | 0.005 | 0.421 | 0.126 | 0.209 | 0.855 |
| SG | Coverage | 0.001 | 0.861 | 0.059 | 0.729 | 0.769 |
| | Plant density | 0.015 | 0.348 | 0.033 | 0.820 | 0.687 |
| | Aboveground biomass | 0.006 | 0.689 | 0.272 | 0.048 | 0.910 |
| LG | Coverage | 0.000 | 0.722 | 0.004 | 0.979 | 0.815 |
| | Plant density | 0.006 | 0.429 | 0.008 | 0.952 | 0.958 |
| | Aboveground biomass | 0.288 | 0.596 | 0.170 | 0.004 | 0.815 |
| FG | Coverage | 0.004 | 0.915 | 0.087 | 0.596 | 0.845 |
| | Plant density | 0.006 | 0.582 | 0.068 | 0.672 | 0.975 |
| | Aboveground biomass | 0.080 | 0.662 | 0.470 | 0.006 | 0.933 |
| NG | Coverage | 0.001 | 0.824 | 0.023 | 0.888 | 0.955 |
| | Plant density | 0.002 | 0.253 | 0.002 | 0.989 | 0.726 |
| | Aboveground biomass | 0.231 | 0.754 | 0.288 | 0.110 | 0.979 |

non-grazed treatment and sampling year on the aboveground biomass of SG, LG and FG was observed (Fig. 3; Table 2).

Non-grazed treatment showed a significant decreasing trends over a period of time on the density of all five vegetation types (Fig. 4; Table 2) displaying minimum inclination in 2017 (Fig. 4; Table 2). In case of grazed treatment, the density of all five vegetation types first decreased and then showed an upward trend during the experimental period (Fig. 4; Table 2) with minimum reading during 2016 (Fig. 4; Table 2). The non-grazed treatment displayed first decreasing and then an upward trend in the aboveground biomass of all five vegetation types (Fig. 4; Table 2) with minimum value occurring during 2016 (Fig. 4; Table 2).

## Vegetation nutritional values response to non-grazed treatment and year variations

In comparison to grazed treatment, non-grazed treatment significantly decreased the CP content of all vegetation types, except NG (Fig. 5; Table 3), it also decreased the IVTD of all vegetation types, except NG (Fig. 5; Table 3). A considerable decrease in the NDF content of all vegetation types, except NG (Fig. 5; Table 3) during non-grazed treatment was also recorded. The sampling year showed positive effect on the CP content of GG and SG (Fig. 5; Table 3). The significant increase in the IVTD of GG and LG were recorded during non-grazed treatment (Fig. 5; Table 3). The NDF content of GG and SG increased were

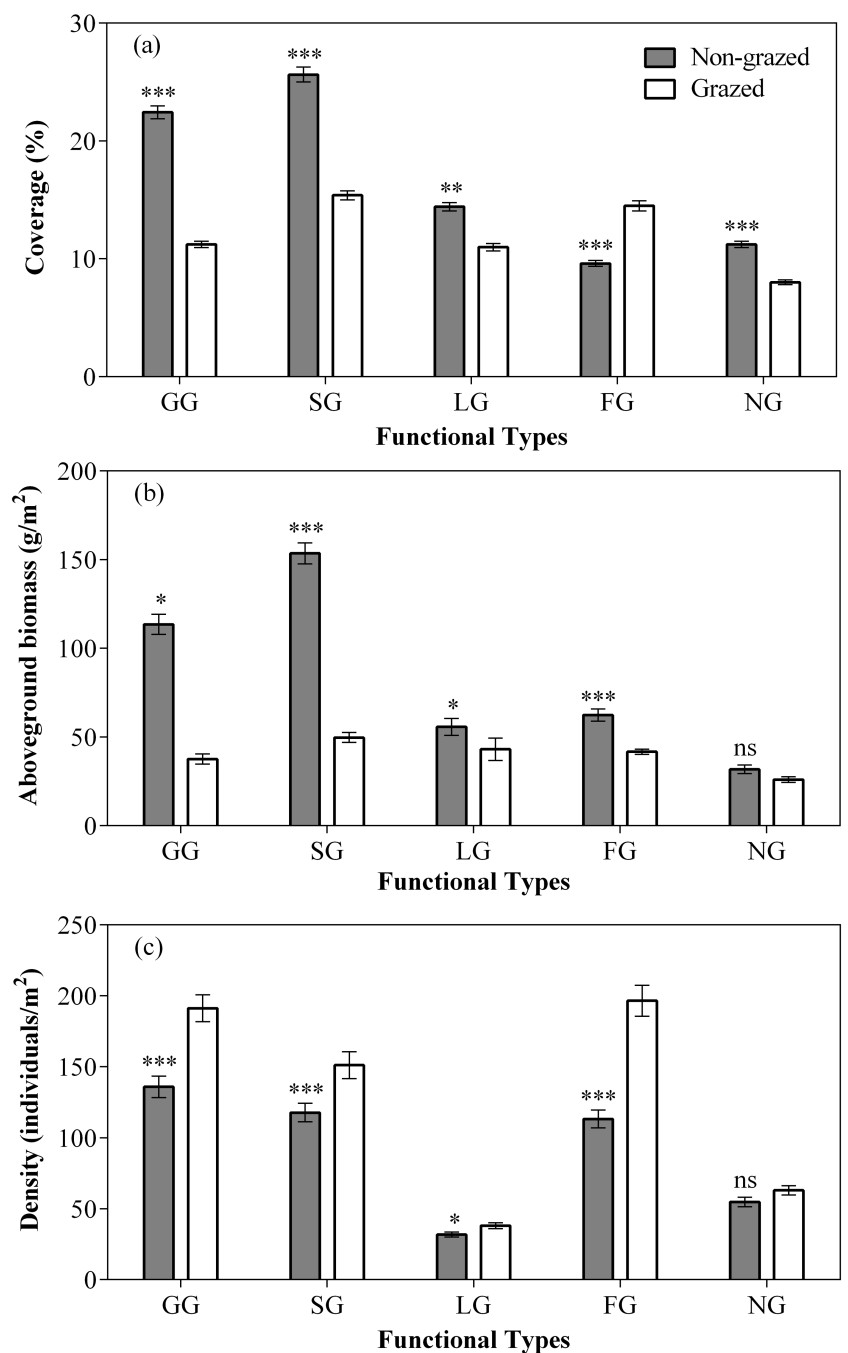

**Figure 3** **Effect of non-grazed and grazed on the coverage (A, %), aboveground biomass (B, g/m²), and density (C, individuals/m²) of five functional types of alpine meadows between non-grazed and grazed treatment.** The values (Mean ± SE) are means of 3 years (2015, 2016 and 2017). For five functional types, GG, SG, LG, FG and NG represents grass species type, sedge species type, leguminous species type, forbs species type and noxious species type respectly. The symbols represent signigicant difference between non-grazed and grazed alpine meadow treatments, ***$P < 0.001$, **$P < 0.01$, *$P < 0.05$; ns, no significant difference.

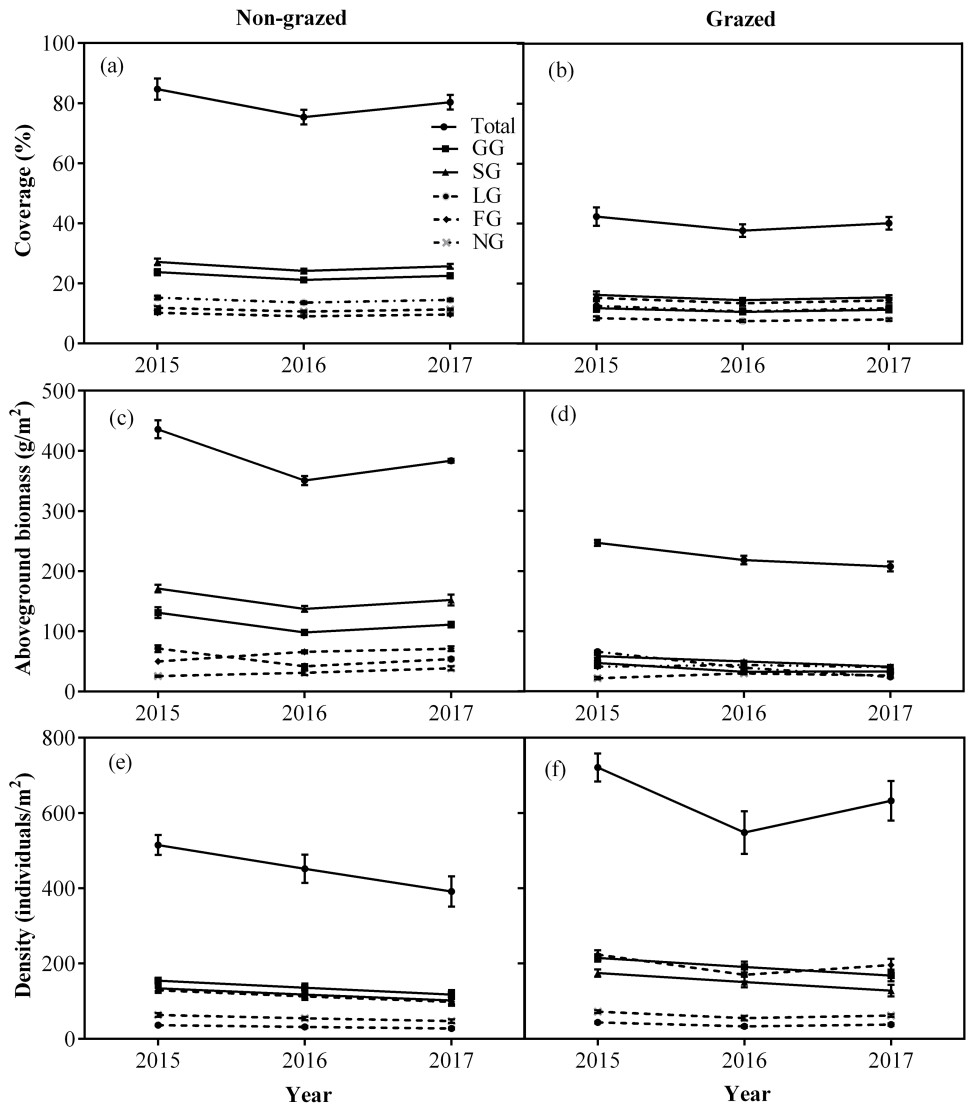

**Figure 4  Changes of coverage (A, %), aboveground biomass (B, g/m$^2$), and density (C, individuals/m$^2$) for total vegetation and five functional types between non-grazed and grazed alpine meadows.** The values (Mean ± SE) are means of 3 years (2015, 2016 and 2017). For five functional types, GG, SG, LG, FG and NG represents grass species type, sedge species type, leguminous species type, forbs species type and noxious species type, respectively.

observed during the sampling year (Fig. 5; Table 3). The relation of non-grazed treatment and sampling year showed significant effect on the CP content of all vegetation types, except NG (Fig. 5; Table 3). While in case of IVTD, only SG showed significant effect (Fig. 5; Table 3). The NDF contents of SG and LG were observed effected during the study of interaction of non-grazed treatment and sampling year (Fig. 5; Table 3) were seen.

The CP and IVTD contents of all vegetation types, except NG in non-grazed alpine meadows showed a gradually decreasing trend (Fig. 6; Table 3) while in case of grazed alpine meadows, the CP and IVTD contents of all vegetation types, except NG firstly

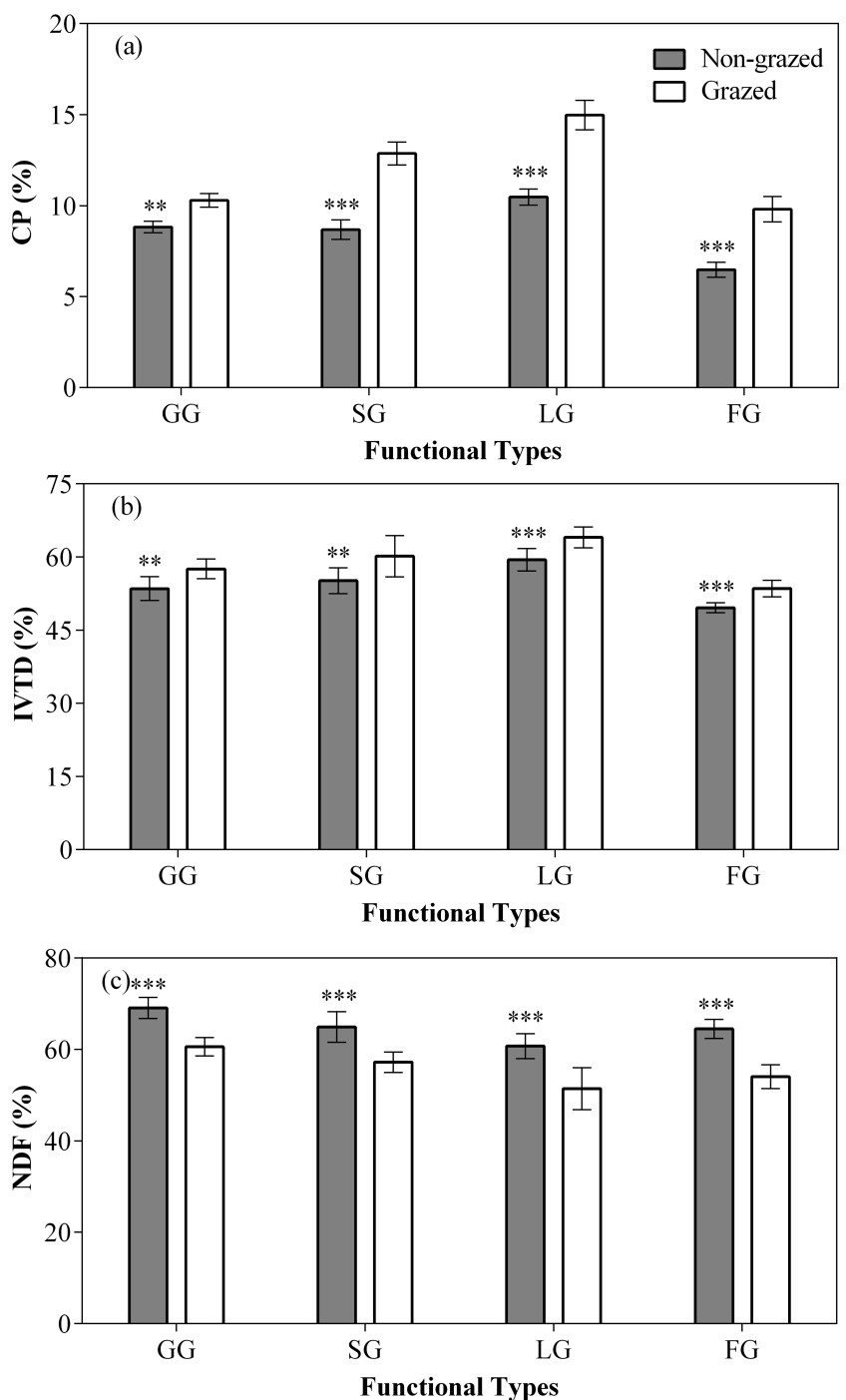

**Figure 5** **Effect of non-grazed and grazed on four edible functional types CP (%), IVTD (%), NDF (%) of alpine meadows between non-grazed and grazed treatment.** The values (Mean ± SE) are means of 3 years (2015, 2016 and 2017); GG, SG, LG and FG represents grass species type, sedge species type, leguminous species type and forbs species type respectly. The symbols represent signigicant difference between non-grazed and grazed alpine meadow treatments, ***$P < 0.001$, **$P < 0.01$, *$P < 0.05$; ns, no significant difference.

**Table 3** The effects of years (2015, 2016 and 2017), non-grazed (comparied with grazed) and interaction between non-grazed and year on CP (%), IVTD (%) and NDF (%) for four edible functional types, GG, SG, LG and FG represents grass species type, sedge species type, leguminous species type and forbs species type respectly.

| Functional types | Items | P-values of variables | | | | |
|---|---|---|---|---|---|---|
| | | Non-grazed | Plots | Year | Non-grazed × Year | Year × Plots |
| | CP | 0.007 | 0.105 | 0.008 | 0.019 | 0.595 |
| GG | IVTD | 0.009 | 0.322 | 0.019 | 0.328 | 0.911 |
| | NDF | 0.004 | 0.122 | 0.035 | 0.279 | 0.859 |
| | CP | 0.005 | 0.230 | 0.015 | 0.000 | 0.871 |
| SG | IVTD | 0.047 | 0.612 | 0.060 | 0.015 | 0.822 |
| | NDF | 0.010 | 0.326 | 0.041 | 0.007 | 0.820 |
| | CP | 0.027 | 0.332 | 0.083 | 0.000 | 0.671 |
| LG | IVTD | 0.001 | 0.235 | 0.001 | 0.904 | 0.902 |
| | NDF | 0.023 | 0.455 | 0.088 | 0.015 | 0.952 |
| | CP | 0.026 | 0.258 | 0.057 | 0.000 | 0.588 |
| FG | IVTD | 0.012 | 0.509 | 0.078 | 0.386 | 0.740 |
| | NDF | 0.020 | 0.311 | 0.227 | 0.000 | 0.574 |

showed decreasing trend and then an upward trend was observed during the experiment period (Fig. 6; Table 3) both showing lowest value during 2016 year (Fig. 6; Table 3). However, the NDF content of all vegetation types, except NG showed an increasing trend in non-grazed alpine meadow during experiment, displaying highest value during year 2017 (Fig. 6; Table 3). In case of grazed alpine meadows, the NDF content of all vegetation types, except NG first increased and then a downward trend was recorded (Fig. 6; Table 3), the highest value was observed in year 2016 (Fig. 6; Table 3).

## Soil properties response to non-grazed treatment and year variations

As compared to grazed treatment, non-grazed treatment significantly increased all six measured soil properties in the 0–10 cm soil layer while an increase in all measured soil properties, except TP were recorded in the 10–20 cm soil layer. In 20–30 cm soil layer all measured soil properties, except TN and TK considerably increased (Fig. 7; Table 4). Sampling year also effected the soil properties such as significant increase in the soil TK, AN and AK in the 0–10 cm soil layer was observed and an increase in the soil AN and AK in the 10–20 cm soil layer (Fig. 7; Table 4) was also recorded. The interaction of non-grazed treatment and sampling year significantly affected the soil TN, TP, AN and AK in the 0–10 cm soil layer. A significant effect on all measured soil properties, except TK in the 10–20 cm soil layer and on all measured soil properties, except TP and TP in the 20–30 cm soil layer were observed (Fig. 7; Table 4).

In non-grazed alpine meadows, the all measured soil properties, except TK of three soil layers i.e., 0–10, 10–20 and 20–30 cm showed a gradually increasing trend during the whole experimental period, the lowest value occurred in 2017 year (Fig. 8; Table 4). While in case of grazed alpine meadows, the TP, TK and AP of three soil layers (0–10, 10–20

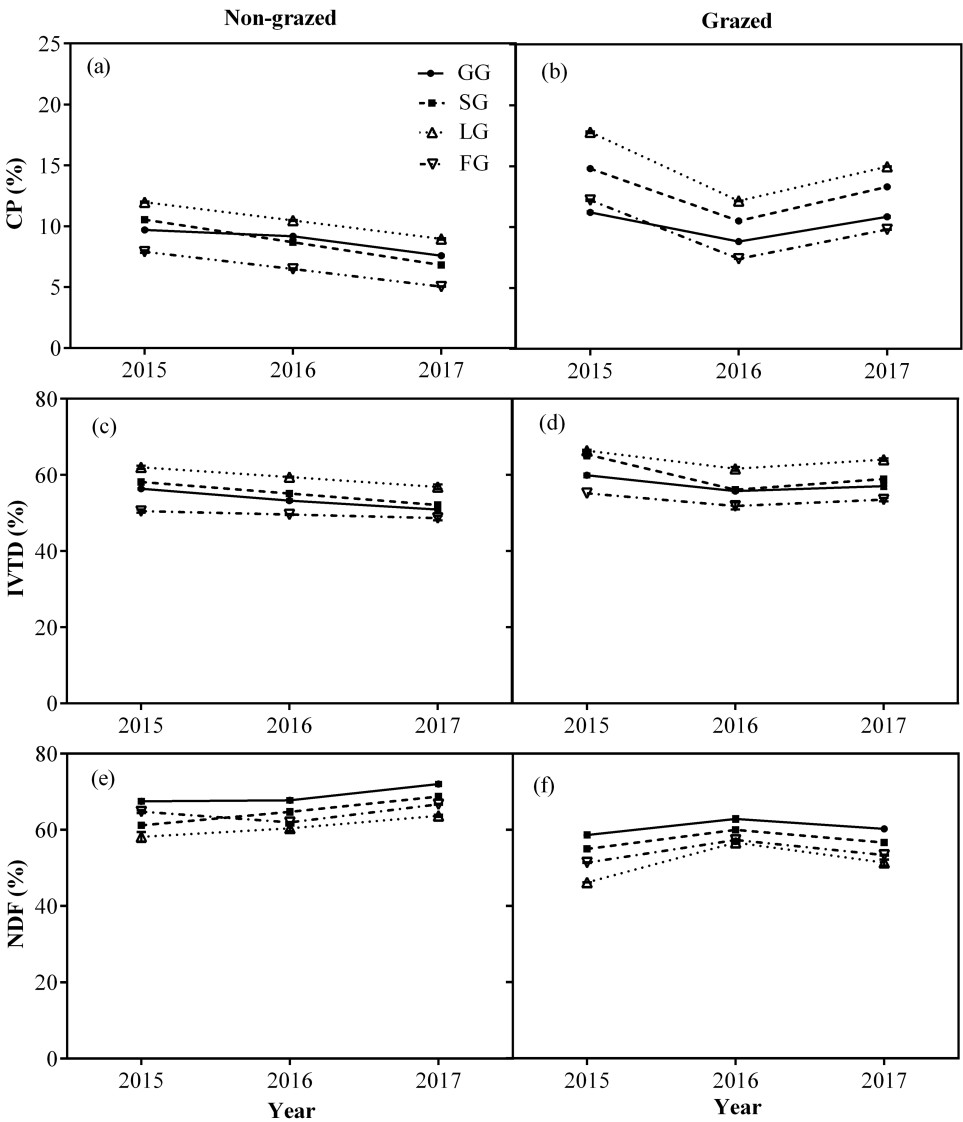

**Figure 6 Changes of CP (%), IVTD (%), NDF (%) for four edible functional types (GG, SG, LG and FG) between non-grazed and grazed alpine meadows.** The values (Mean ± SE) are means of 3 years (2015, 2016 and 2017); GG, SG, LG and FG represents grass species type, sedge species type, leguminous species type and forbs species type, respectively.

and 20–30 cm) first decreased and then an upward trend was recorded during the whole experimental period with the lowest value displayed in 2016 (Fig. 8; Table 4). Overall TN, AN and AK of three soil layers (0–10, 10–20 and 20–30 cm) showed a gradually decreasing trend during the whole experimental period, and displayed the lowest value in 2017 year (Fig. 8; Table 4).

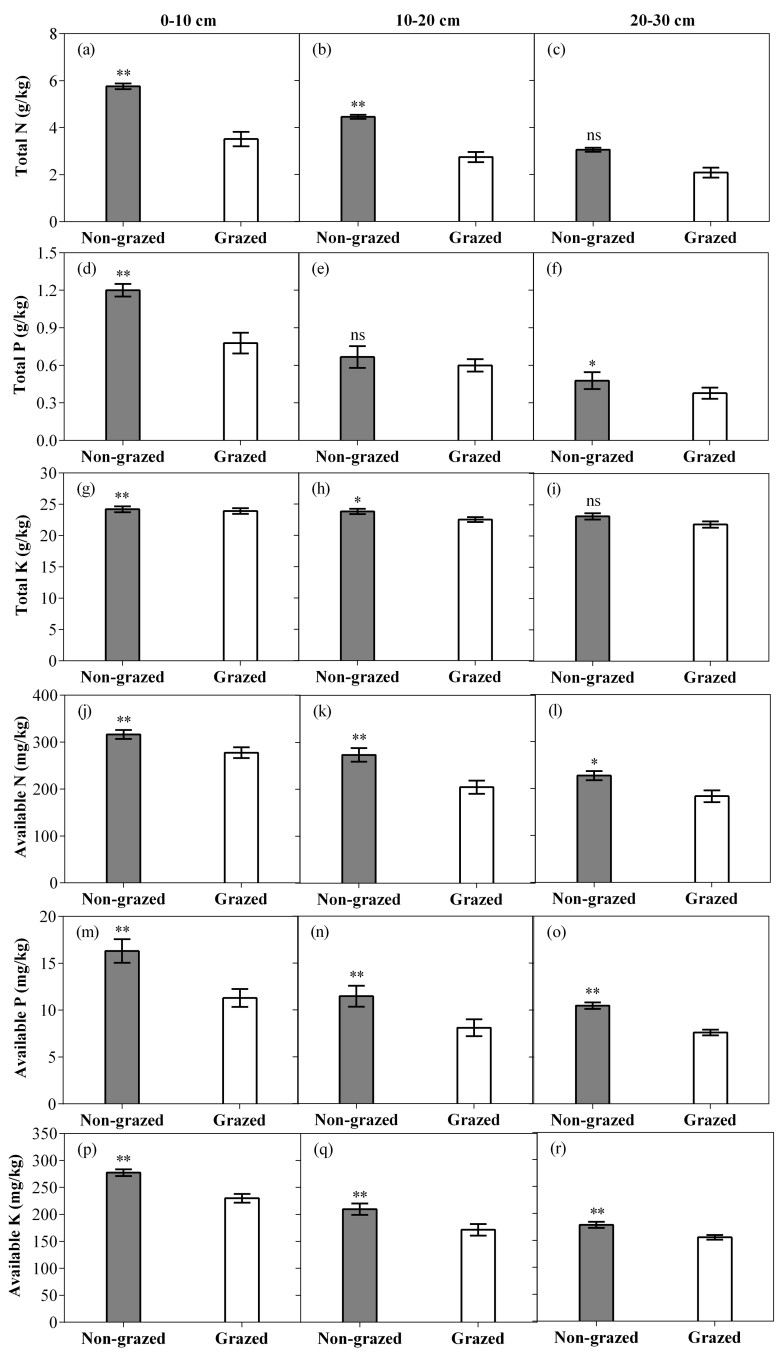

**Figure 7 Effect of non-grazed and grazed on soil properties TN (g/kg), TP (g/kg), TK (g/kg), AN (mg/kg), AP (mg/kg) and AK (mg/kg) of alpine meadow between non-grazed and grazed treatment in 3 (0–10, 10–20 and 20–30 cm) soil depths.** The values (Mean ± SE) are means of 3 years (2015, 2016 and 2017). The symbols represent signigicant difference between non-grazed and grazed alpine meadow treatments, $***P < 0.001$, $**P < 0.01$, $*P < 0.05$; ns, no significant difference.

Table 4 The effects of years (2015, 2016 and 2017), non-grazed (compared with grazed) and interaction between non-grazed and year on TN, TP, TK, AN, AP and AK for 3 (0–10, 10–20 and 20–30 cm) soil depths.

| Soil properties | Soil depth (cm) | P-values of variables | | | | |
|---|---|---|---|---|---|---|
| | | Non-grazed | Plots | Year | Non-grazed × Year | Year × Plots |
| Total N (g/kg) | 0–10 | 0.004 | 0.364 | 0.247 | 0.002 | 0.856 |
| | 10–20 | 0.003 | 0.662 | 0.252 | 0.006 | 0.854 |
| | 20–30 | 0.018 | 0.522 | 0.419 | 0.001 | 0.815 |
| Total P (g/kg) | 0–10 | 0.006 | 0.357 | 0.258 | 0.020 | 0.911 |
| | 10–20 | 0.151 | 0.647 | 0.241 | 0.019 | 0.847 |
| | 20–30 | 0.030 | 0.522 | 0.291 | 0.232 | 0.996 |
| Total K (g/kg) | 0–10 | 0.007 | 0.536 | 0.002 | 0.948 | 0.995 |
| | 10–20 | 0.042 | 0.541 | 0.285 | 0.011 | 0.866 |
| | 20–30 | 0.072 | 0.611 | 0.377 | 0.001 | 0.851 |
| Available N (mg/kg) | 0–10 | 0.002 | 0.233 | 0.015 | 0.001 | 0.634 |
| | 10–20 | 0.001 | 0.416 | 0.018 | 0.000 | 0.513 |
| | 20–30 | 0.023 | 0.411 | 0.205 | 0.000 | 0.715 |
| Available P (mg/kg) | 0–10 | 0.005 | 0.366 | 0.128 | 0.595 | 0.978 |
| | 10–20 | 0.017 | 0.622 | 0.363 | 0.438 | 0.710 |
| | 20–30 | 0.020 | 0.378 | 0.494 | 0.480 | 0.814 |
| Available K (mg/kg) | 0–10 | 0.001 | 0.255 | 0.019 | 0.006 | 0.456 |
| | 10–20 | 0.001 | 0.425 | 0.005 | 0.037 | 0.642 |
| | 20–30 | 0.012 | 0.291 | 0.152 | 0.000 | 0.914 |

## DISCUSSION

### Vegetation characteristics response to non-grazed treatment and year variations

The restoration of degraded grassland ecosystem is a complex and long-term ecological process (*Cheng et al., 2016*; *Török & Helm, 2017*; *Török et al., 2018*). Non-grazed treatment is generally regarded as a useful tool to restore the productivity of degraded grasslands ecosystem (*Spooner, Lunt & Robinson, 2002*). Many studies have shown that the restoration of grassland by non-grazed grasslands can be assessed by biomass, coverage, density and diversity of vegetation (*Wilkins, Keith & Adam, 2003*; *Cheng et al., 2016*). In our research, we selected aboveground biomass, coverage, density, biodiversity and richness to assess the impact of non-grazed treatment on vegetation. Research has shown that climatic factors are also the main driving force of degradation, greater than overgrazing for alpine grassland (*Niu, Ma & Zeng, 2008*). The variation of vegetation biomass between years is primarily influenced by local rainfall, temperature and sunshine radiation (*Akiyama & Kawamura, 2007*; *Niu, Ma & Zeng, 2008*; *Wu et al., 2009*; *Miao et al., 2015*; *Ren et al., 2016*). In our study area, the annual average temperature and ≥0 °C accumulative temperature gradually increased, while annual rainfall decreased (Fig. 1). That indicates that there was a warmer and dry trend in the local climate, which might be a very influential factor in the succession of plant communities (*Bai et al., 2004*).

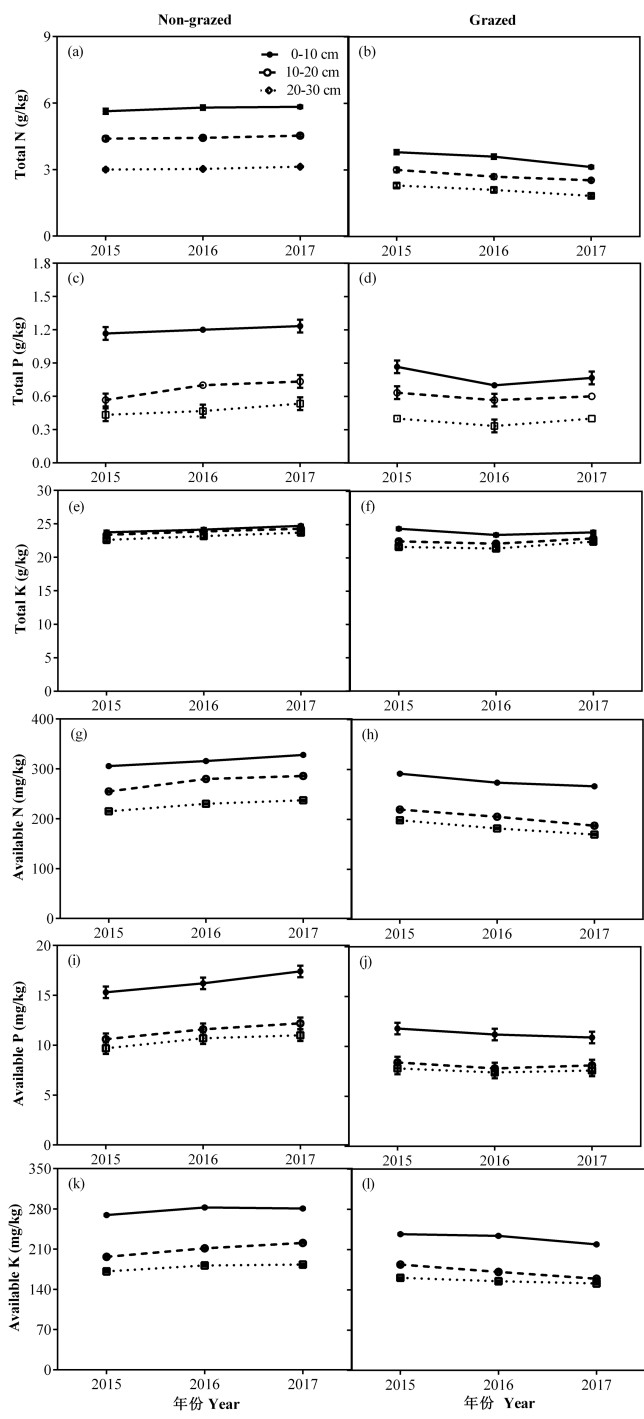

**Figure 8   Changes of TN (g/kg), TP (g/kg), TK (g/kg), AN (mg/kg), AP (mg/kg) and AK (mg/kg) for three (0–10, 10–20 and 20–30 cm) soil depths between non-grazed and grazed alpine meadows.** The values (Mean ± SE) are means of 3 years (2015, 2016 and 2017).

Our study showed that long-term non-grazed exclosures remarkably increased the aboveground biomass and coverage of vegetation, but decreased the density and biodiversity of vegetation (Fig. 2; Tables 1 and 2). Similar results have also been reported in other grassland types (*Haugland & Froud-Williams, 1999*; *Wu et al., 2009*; *Wang et al., 2012*). Non-grazed exclosures increased the coverage and aboveground biomass of gramineous (GG) and sedge (SG) plants in alpine meadow communities by excluding intake of herbivores, which had good palatability for domestic animals (Figs. 3 and 4; Tables 1 and 2). Studies have shown that forage with good palatability is more competitive than those forage with poor palatability, and non-grazed exclosures significantly increased the biomass of gramineous and sedge species which have good palatability (*Gallego et al., 2004*; *Wu et al., 2009*). These results are consistent with previous reports (*Gallego et al., 2004*; *Wu et al., 2009*; *Shang et al., 2013*), supporting that non-grazed exclosures benefits for the improvement of biomass and coverage of four plant functional types (GG, SG, LG and FG) and inhibition in the development of noxious species type (NG). Livestock grazing accelerated the loss of plant roots and leaf biomass, promoted the recycling of nutrients (*Semmartin, Garibaldi & Chaneton, 2008*), and decreased the vegetation biomass of grazed alpine meadows. However, non-grazed exclosures showed a negative effect on the density and biodiversity of plant functional types. In high biomass grasslands, the loss of vegetation diversity might be due to greater competition of canopy resources (i.e., light and air) (*Huston, 1994*). Some of the less competitive species had limited availability of light or nutrient (*Grime, 1998*; *Van Der Wal et al., 2004*), resulted in their decrease density or species disappearance. On the contrary, grazed treatment decreased the biomass of dominant functional types (GG and SG) (Table 1), allowing other functional types in the community to have more development opportunities and promoting balanced development of the community.

The alpine meadows 11–13 years non-grazed exclosures reflected changes in functional types from smaller to larger, and in density from higher to lower (Table 1). In non-grazed alpine meadows, the biodiversity of five plant functional types (GG, SG, LG, FG and NG) decreased (Table 1). The density changes of five plant functional types (GG, SG, LG, FG and NG) in the non-grazed area means that the number of species in the non-grazed alpine meadow was less, total aboveground biomass was higher and number of newly appeared species is fewer (Figs. 2 and 3; Table 1). This indicated that the non-grazed exclosures has affected the concealment of the habitat and led to the loss of species diversity. The concealment of the habitat determines the supplement of local plant seedlings (*Oba, Vetaas & Stenseth, 2001*) and disruption of some unusual plant species (*Inderjit, 2005*). The species density and diversity in the fenced alpine meadow is significantly higher than grazing alpine meadow. Grazed treatment inhibits the development of dominant community (GG and SG), increases spatial heterogeneity, and makes other functional types (LG, FG and NG) in the community to grow to achieve a balanced development (Figs. 3 and 4), which is consistent with the previous reports (*Begon, Harper & Townsend, 1990*; *Sheppard et al., 2002*; *Schippers & Joenje, 2002*; *Holdo et al., 2007*; *Wu et al., 2009*).

Grazed treatment is regarded as a key factor leading to grassland degradation; meanwhile it is also a main driving force for grassland succession (*Holdo et al., 2007*). Plant diversity is

mainly dependent on grazing intensity. Overgrazing may lead to the grassland degradation and biodiversity loss and light grazing may lead to grassland succession to woodland and the loss of grassland habitats. Not only is grazed intensity important, but the time of grazing and the type of grazed livestock are also important (*Hulme et al., 1999*). It is necessary to do more research about the effects of non-grazed and grazed on alpine meadows, especially in terms of global climate change (*Watkinson & Ormerod, 2001*).

## Vegetation nutritional values response to non-grazed treatment and year variations

It has been reported that grazing usually increased the forage nutritional value (*Bai et al., 2012*; *Schönbach et al., 2012*), which will in turn affects the performance of livestock (*Mysterud et al., 2001*; *Mysterud et al., 2011*; *Lin et al., 2011*; *Müller et al., 2014*). Our research showed that grazed treatment significantly increased the nutritional value (CP and IVTD) of four edible functional types (GG, SG, LG and FG), which is consistent with other studies (*Schönbach et al., 2009*; *Fanselow et al., 2011*; *Ren et al., 2016*). Firstly, a large amount of nitrogen was stored in the stems and leaves, because grazed forage had a higher relative absorbance of nitrogen and it was moved into the young tissue of shoots and leaves when herbage was taken (*Lambers et al., 2009*; *Fanselow et al., 2011*). Secondly, nitrogen originating from the dung and urine of grazed livestock had a positive effect on nitrogen concentrations of forage and herbivore excretions usually accelerated the rate of mineralization of the soil surface by senescent plant litter. As the soil mineral nitrogen increased, the nitrogen content in plants also increased (*Semmartin, Garibaldi & Chaneton, 2008*; *Wang et al., 2009*; *Jiang et al., 2012*; *Miao et al., 2015*). Finally, grazed treatment could affect the nutritional value of plants by using young and protein-rich parts and regenerating it, instead of the aged parts of plants (*Mysterud et al., 2001*; *Schönbach et al., 2009*; *Ren et al., 2016*). Due to the regeneration of new tissues under grazing pressure, the maturation and lignification of the species are delayed, and the CP content increased; our findings support these results (*Milchunas et al., 1995*; *Garcia et al., 2003*).

The CP and IVTD of four edible functional types (GG, SG, LG and FG) showed a gradually decreasing trend with the increase of non-grazed time, while NDF gradually increased. The CP and IVTD of functional types in grazed alpine meadows showed a decreasing trend at first and then increasing trend with time was observed. Contrary to the changing trend of CP and IVTD, NDF content showed at first rising trend and then decreasing trend. This might be due to rainfall in the study area. As reported previously that variation in precipitation rate between years affects the herbage nutritional values (*Miao et al., 2015*). The herbage nutritional value depends on the amount of precipitation, which increases with increasing rainfall, consistent with past research (*Schönbach et al., 2009*; *Müller et al., 2014*; *Miao et al., 2015*). Our results also indicated that nutritional value of four edible functional types in relatively wet years (2015 and 2017) was higher than in dry year (2016) (Fig. 1). Plant growth was limited by the amount of precipitation and water availability (*Miao et al., 2015*). In relatively wet years, abundant precipitation accelerated soil water utilization rate and soil mineralization, and promoted the ability of plants to absorb nitrogen (N), thus promoting biomass production (*Austin et al., 2004*; *Xu & Zhou,*

*2005*). Meanwhile, drought can caused severe water stress, resulted in rapid ripening of plants, thereby reducing the concentration of N in forage. Therefore, water stress in drought years increases forage fibrosis and reduces forages digestibility, while in wet years due to high rainfall maturation process delays, forage fibrosis reduces and in turn forage digestibility was improved. However, in the case of non-grazed alpine meadow, the CP and IVTD of four edible functional types (GG, SG, LG, and FG) did not show any increasing trend with increase in rainfall. This may be due to the fact that as the non-grazed time was increased, biomass accumulated with the passage of every year. Mature and aged tissues inhibited the germination and growth of young tissues from the seedlings in the growing season, causing the CP and IVTD to decrease and NDF to increase every year.

Livestock grazing can significantly change the structure and nutritional value of vegetation and their trampling behavior and excrement can also affect the community structure and soil properties of the ground (*Gibson et al., 2000*). Therefore, vegetation succession, functional types characteristics and nutritional values are closely related to livestock grazed. For the succession of grasslands vegetation and utilization of grassland resources, regular grazed and non-grazed treatment are beneficial to grasslands management.

**Soil properties response to non-grazed treatment and year variations**
As the non-grazed time increased, the soil TN, TP, AN, AP and AK went significantly up, showing that the soil nutrients of degraded alpine meadow were being restored by fencing approach (*Jing et al., 2014*), indicating that natural succession of degraded soils in alpine meadow areas of the QTP could improve soil fertility. The improvement of soil properties in alpine meadows with increased of non-grazed time had two explanations: first, the productivity of vegetation has a direct impact on the accumulation of litter. With the accumulation of litter and in the presence of soil moisture, litter decomposition rate enhanced and soil nutrients showed an increasing trend (*Wu et al., 2009*). Secondly, higher soil nutrients might be due to higher community coverage. Previous studies have found that vegetation coverage has an obvious impact on the quality of soil nutrients (*Zhang et al., 2011*), our results further supports these findings.

The interaction of soil and plant is a complex process (*Lambers et al., 2009*). The movement of energy and nutrients in soil can directly and indirectly reflect the species composition, productivity and nutritional value of vegetation (*Venterink, 2011*). In non-grazed grasslands, firstly, the plant community of non-grazed grassland locked-in nutrients (*Harris et al., 2007*) in their tissues, reduced the outflow of energy and nutrients from soil-plant system to the consumer (grazed livestock), especially gramineous (GG) and sedge (SG) functional types had high productivity and good quality (*Moretto & Distel, 1997*). Vegetation resources (coverage and productivity) were significantly improved with the increase in non-grazed time. These resources could go back to the soil by the decomposition of the litter layer (*Bardgett & Wardle, 2003*; *Wu et al., 2009*). Secondly, non-grazed removed the trampling effect of grazing livestock, improved soil characteristics, increased water interception and improved vegetation status (*Li et al., 2007*). Along with the development of aboveground vegetation, the better vegetation conditions reduced

the wind erosion and some nutrients richer particles and dust was captured in the soil (*Liu et al., 2007*). Thirdly, the improvement of soil nutrients had a positive regeneration effect on the aboveground biomass and structure of plant functional types, because the utilization of higher nutrient levels is beneficial for the competition of gramineous (GG) and sedge (SG) functional types to other species (*Van Der Wal et al., 2004*). Finally, decrease in the quantity of rodents (*Myospalax fontanierii* and *Microtus leucurus*) has a positive effect on soil biological communities and soil processes by altering the input of soil resources (*Bardgett & Wardle, 2003*). However, for grazed grasslands, some energy and nutrients flow from soil-plant system to livestock by grazing, which at first altered soil properties, reduced the litter and root biomass that were fed back into the soil after decomposition (*Gao et al., 2008*); Secondly, the edible functional types grazed by livestock had higher litter decomposition rate and efficient soil nitrogen than inedible functional group forage (*Moretto & Distel, 2002*). Finally, long-term trampling by livestock transforms soil composition, infiltration rates, bulk density, soil porosity, limiting soil respiration and reducing soil microbial activity (*Holt, 1997*).

Therefore, soil nutrient has a positive effect on the aboveground biomass, composition and nutritional value of plant functional types. However, this study only analyzed the soil chemical characteristic. In future, the relationship between plant functional types and soil physics or biology should also be considered.

## CONCLUSIONS

The restoration of a degraded grassland ecosystem is a complex long-term ecological process (*Lambers et al., 2009*). Eleven to thirteen years of non-grazed exclosures in QTP have led to subsequent changes of plant functional type biomass, structure, nutritional values, quantity and quality of litter inputs to the soil. Long-term non-grazed exclosures have increased aboveground biomass and coverage of plant functional types. It is beneficial for the improvement of four edible functional types (GG, SG; LG and FG), but inhibited the development of the NG type. Long-term non-grazed exclosures also significantly improved 0–30 cm soil TN, TP, TK, AN, AP and AK. However, it decreased the species biodiversity indicators, including the species richness, the Shannon diversity index and the Evenness index of vegetation. It also decreased the density, biodiversity and nutritional value of four edible plant functional types (GG, SG, LG and FG). There exist a dilemma between biodiversity protection and grazed utilization in grasslands under heavy grazing pressure and long-term non-grazed exclosures. As disturbance measures, non-grazed and grazed treatment had opposite impacts on aboveground biomass, coverage, density, biodiversity, nutritional values and soil properties. However, it was from the highest to moderate levels of disturbance for species density and diversity, according to the ''intermediate disturbance hypothesis'' (*Connell & Slatyer, 1977*). Landsberg reported that moderate grazing pressure increases the diversity of vegetation at local natural biotope (*Landsberg et al., 2002*). Our research indicated that non-grazed exclosures can be used as a useful restoration tool implemented at large scales in many regions to restore aboveground biomass and coverage of degraded grassland. Meanwhile, grazed treatment could be used as

an effective grasslands management strategy to increase biodiversity and nutritional values of plant functional types in long-term fenced grasslands. We recommend that long-term non-grazed grasslands should reasonably utilize non-grazed and grazed treatment for grassland management. We suggest that rotational grazed and non-grazed treatment can be regarded as a beneficial management strategy for grasslands management around the world where common problems exist. More meaningful studies should be carried out in the future for the restoration, management and utilization of grassland, such as fertilization, fencing time, grazing intensity, grazing time and grazed livestock species.

## ACKNOWLEDGEMENTS

We acknowledge the herdsmen and Sunan County Meteorological Bureau for their assistance and for providing long-term meteorological data.

### Funding

This study was funded by the National Natural Science Foundation of China (#31460592), the China's Agricultural Research system (CARS-39-18) and a research project on public welfare industry (Agriculture) from the Ministry of Agriculture (201503134-HY15038488). The funders had no role in study design, data collection and analysis, decision to publish, or preparation of the manuscript.

### Grant Disclosures

The following grant information was disclosed by the authors:
National Natural Science Foundation of China: #31460592.
China's Agricultural Research system: CARS-39-18.
Research project on public welfare industry (Agriculture) of the Ministry of Agriculture: 201503134-HY15038488.

### Competing Interests

The authors declare there are no competing interests.

### Author Contributions

- Xixi Yao conceived and designed the experiments, performed the experiments, analyzed the data, contributed reagents/materials/analysis tools, prepared figures and/or tables, authored or reviewed drafts of the paper, approved the final draft.
- Jianping Wu conceived and designed the experiments, performed the experiments, authored or reviewed drafts of the paper, approved the final draft.
- Xuyin Gong performed the experiments, analyzed the data, prepared figures and/or tables, authored or reviewed drafts of the paper, approved the final draft.
- Xia Lang contributed reagents/materials/analysis tools, approved the final draft.
- Cailian Wang prepared figures and/or tables, approved the final draft.

## Field Study Permissions

The following information was supplied relating to field study approvals (i.e., approving body and any reference numbers):

The College of Animal Science and Technology of Gansu Agricultural University approved the study.

## Data Availability

The raw data is provided in Table 1.

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
