# Peer review of "Grazing exclosures solely are not the best methods for sustaining alpine grasslands"

_PeerJ, doi:10.7717/peerj.6462_

## Round 0.1 · original submission · Major Revisions

Dear authors,

I have received reviews from two expert reviewers. To help us assess your action in response to their insightful reviews please provide fully annotated response to all of their detailed comments. To help you do this I draw your attention to several technical issues: please address the over use of abbreviations in the abstract. Consider the use of non-grazed and grazed terms for the two treatments. Address the need to be more succinct in the Introduction and Discussion as suggested by R1 and address the repetitive nature of Results presentation highlighted by R2.

Regards
Steve

Reviewer 1 ·

Basic reporting

The manuscript reports results from a grazing exclusion study conducted in alpine grasslands in China. The authors aimed at to study grazing effect on overall diversity and predefined groups of plant species using line transects and plot-based sampling. The ms in general well written, however, some parts are too voluminous and wordy (i.e. introduction and discussion). The main finding of the ms is that neither continuous grazing, nor exclosures can provide the desired biodiversity of grasslands thus, they suggested an alternative method for some kind of mosaic management, where exclosures and grazed sites alternate both in space and time. The figures and illustrations are appropriate for the study. The manuscript is generally well written and clear.

Experimental design

The study setting is appropriate, however the distance between the different treatments/sites are rather low and falls between the general spontaneous dispersal distance of grassland plants (especially if the dispersal is facilitated by grazing). It cannot be treated in revision but maybe its effects can be discussed briefly in methods (i.e. which drivers motivated the close sampling sites). The statistical analyses are appropriate, however, I don’t know why the authors used two types of multiple comparasions for the analyses (i.e. the default setting of GLMM is LSD but the authors also stressed that they have calculated Tukey). I also think that ‘noxius species’ is not a functional group – it can contain in general many plants species with different plant strategies but unwanted in the sites (e.g. weeds, invasive species mostly grasses and forbs)

Validity of the findings

The results are solid, clear and are in line with other studies. Relatively detailed analyses support the findings.

Additional comments

I have identified a set of minor issues which should be treated in a revision

Title – is rather overloaded (it is not the abstract!) should be more focused on the most important findings – i.e. grazing exclosures solely are not the best methods for sustaining alpine grasslands - a case study from China – or smthing similar.

l35 noxius species is not a functional group in ecological terms surely
l39 Richness index – it is just species richness – why you don’t write it?
l54-57 Some very recent overviews should be considered here – mostly focusing on grasslands and steppes

Török P., Dengler J. (2018): Palaearctic grasslands in transition: overarching patterns and future prospects. In: Squires, V.R., Dengler, J., Feng, H., Hua, L. (eds.) Grasslands of the world: diversity, management and conservation. CRC Press, Boca Raton, US. Ch. 2. pp. 15-26.

Wesche K., Ambarli D., Kamp J., Török P., Treiber J., Dengler J. (2016): The Palaearctic steppe biome: a new synthesis. Biodiversity & Conservation 25: 2197-2231.

Török P., Wesche K., Ambarli D., Kamp J., Dengler, J. (2016): Step(pe) up! Raising the profile of the Palaearctic natural grasslands. Biodiversity & Conservation 25: 2187-2195.

l102 Thus, specific research should be…
l132 In order to better understand…
l168 Should be recalculated to Livestock units/ ha LU/ha
l179 Do not understand the sentence – have you sampled the grasslands at the peak of biomass production?
l180 Not “quadrants” – quadrats or plots should be written here and should be corrected in the whole ms
l181 Not sure that it is called “marginal effects” – effect of margin would be better
l184 five groups that included
l194 Simply Shannon diversity – Wiener has nothing to do with it. See original publication of Shannon (1948) there is only one author.
l240-264 Almost impossible to read, so many p values – it should be put in a table.
for the line 266-291 is almost the same.
l317 Quite recent restoration literature should be cited here – Török & Helm 2017 and/or Török et al. (2018)

Török P., Helm A. (2017): Ecological theory provides strong support for habitat restoration. Biological Conservation 206: 85-91.

Török P., Helm A., Kiehl K., Buisson E., Valkó O. (2018): Beyond the species pool: modification of species dispersal, establishment, and assembly by habitat restoration. Restoration Ecology 26: S65-S72.

l366 typo – The species density and diversity
l390 there is no “vegetation community” – There is either “vegetation” or “plant community”. This case I would prefer the first version – please correct consecutively in the ms.

Reviewer 2 ·

Basic reporting

.

Experimental design

.

Validity of the findings

.

Additional comments

In my opinion authors provide a useful contribution to knowledge about the functioning of pastures, so their work can expect a broad interest from scientist working with this topic. However there are a lot of problems with the structure and the style of the manuscript. There are a lot of abbreviations in the abstract; therefore it is impossible to follow the story line. The denomination of the two studied pasture types (e.g. fenced and grazed) is misleading, because the grazed sites were also fenced allowing the prescribed grazing. I suggest the non-grazed and grazed phrasing. Authors studied the effect of grazing on the edible and on the noxious species, which is an important part of the manuscript; however there are no any information about the classification methods. The result section is quite long mostly because of the lot of repetitive information; it is typical that one result was presented in three forms (text, table and figure); I recommend you shorten the length of this part. The labelling and the legend of Fig. 1 are not consistent. The names of the subspecies in Table 1 do not meet the standards of the scientific nomenclature. Moreover, there are several linguistic problems which require the linguistic correction of the whole manuscript. Overall, the topic of the manuscript is interesting and the conservation relevance of these kinds of studies is undisputed. Besides of the ecological viewpoints, Authors concluded their results also from the practical viewpoint. In my opinion, the manuscript is acceptable, however, because of the several technical problems; the thorough re-write of the whole manuscript is highly required.

---

## Round 0.2 · Minor Revisions

Dear Authors,

Many thanks for a detailed and considered response to all the reviewers comments. You should be commended on making it easy to track the changes and improvements in the manuscript.

However, I would request that you further consider changes in the Results section to improve "readability" by finding ways to further reduce the recurrent presence of acronyms. I have rewritten the first paragraph of the results section to demonstrate what I mean. This simple approach replaces five acronyms with the phrase 'all five vegetation types', and when there are four acronyms presented, it replaces it with the phrase 'all vegetation types, except ??'.

Please consider this and then apply this approach to the rest of the results section:

"In comparison to grazed treatment, non-grazed treatment considerably increased the coverage of all five vegetation types (Fig. 3; Table 2). Significant increase in the aboveground biomass of GG, SG was observed during non-grazed alpine meadows. Non-grazed treatment also increased the density of all five vegetation types (Fig. 3; Table 2). The sampling year only increased the coverage of LG and NG. The sampling year also increased the density of all vegetation types, except FG. However, a significant interaction effect between non-grazed treatment and sampling year on the aboveground biomass of SG, LG and FG was observed (Fig. 3; Table 2)."

A similar approach can be taken when describing the results of soil properties. There are five soil properties measured. When all five show a similar response or change, you do not need to write all five acronyms, you can simply state that 'all five measured soil properties' changed....

By looking for these opportunities to reduce the use of acronyms makes the manuscript a far more accessible and satisfying read.

You can also consider this in minor revisions of the abstract as there are still many repeated acronyms there also.

Line 201 - briefly describe how the five soil analyses were done. You cannot simply state they 'were done'.

If you are able to attend to this minor revisions satisfactorily, this manuscript will be ready for publication.

Regards
Steve

---

## Round 0.3 · accepted · Accept

Dear authors,

Many thanks for attending to those last requests for simplification of results description, and greater detail on some methods used.
I am happy to make an 'Accept' recommendation.

Regards
Steve

#